# Broadly Neutralizing Antibodies for Influenza: Passive Immunotherapy and Intranasal Vaccination

**DOI:** 10.3390/vaccines8030424

**Published:** 2020-07-29

**Authors:** Mrityunjoy Biswas, Tatsuya Yamazaki, Joe Chiba, Sachiko Akashi-Takamura

**Affiliations:** 1Department of Microbiology and Immunology, School of Medicine, Aichi Medical University, Aichi 480-1195, Japan; mrityunbio@gmail.com (M.B.); sachiko@aichi-med-u.ac.jp (S.A.-T.); 2Department of Biological Science and Technology, Tokyo University of Science, Tokyo 125-8585, Japan; chibaj@rs.noda.tus.ac.jp

**Keywords:** cross protection, influenza virus, broadly neutralizing mAb (bnAb), camelid variable domain of heavy chain only antibody (V_H_H), multidomain antibody (MDAb), passive immunization, intranasal vaccination, polymeric IgA (pIgA)

## Abstract

Influenza viruses cause annual epidemics and occasional pandemics. The high diversity of viral envelope proteins permits viruses to escape host immunity. Therefore, the development of a universal vaccine and broadly neutralizing antibodies (bnAbs) is essential for controlling various mutant viruses. Here, we review some potentially valuable bnAbs for influenza; one is a novel passive immunotherapy using a variable domain of heavy chain-only antibody (V_H_H), and the other is polymeric immunoglobulin A (pIgA) induced by intranasal vaccination. Recently, it was reported that a tetravalent multidomain antibody (MDAb) was developed by genetic fusion of four V_H_Hs, which are bnAbs against the influenza A or B viruses. The transfer of a gene encoding the MDAb–Fc fusion protein provided cross-protection against both influenza A and B viruses in vivo. An intranasal universal influenza vaccine, which can induce neutralizing pIgAs in the upper respiratory tract, is currently undergoing clinical studies. A recent study has revealed that tetrameric IgAs formed in nasal mucosa are more broadly protective against influenza than the monomeric and dimeric forms. These broadly neutralizing antibodies have high potential to control the currently circulating influenza virus.

## 1. Introduction

Cross protection against infectious diseases is extremely important due to the existence of multiple virus subtypes. Vaccines have successfully eradicated smallpox and can control other viral infections [1]. However, not all current vaccines are universal. For example, seasonal viruses such as influenza spread around the world because antigenic drift and antigenic shift, mainly of membrane proteins such as hemagglutinin (HA), permit the virus to escape host immunity [2,3]. There are fewer therapeutic monoclonal antibodies (mAbs) against viral infections than there are against cancer and autoimmune disease [4,5], mainly because viral antigens are continuously evolving [6]. Therefore, the development of a universal vaccine or broadly neutralizing mAbs (bnAbs) is important to assist with the control of viral infections.

Here, we review how cross-protection can be achieved, via passive immunotherapy or vaccination, against influenza, a representative viral infection. Recently, several bnAbs against the influenza virus that produce cross-protection have been isolated [7]. First, we focus on the engineering of novel antibodies, such as multidomain antibodies (MDAbs), which link camelid single-domain antibodies to protect against influenza A and B infections. It has been reported that an MDAb neutralized both influenza A and B viruses [8]. Current passive immunotherapy using MDAbs may provide broad protection against numerous virus infections. It is well known that influenza viruses invade the respiratory tract, where polymeric IgAs (pIgAs), which play a key protective role against the virus, are produced [9]. Second, we focus on intranasal vaccines that induce pIgA generation in the upper respiratory mucosa and protect against influenza virus infection [9,10]. Recently, it was revealed that tetrameric IgA has better neutralizing activity than monomeric or dimeric IgAs [11].

## 2. Antibody Engineering for Cross Protection against Influenza Virus Infection

### 2.1. Structural and Physiological Features of HA as a Therapeutic Target

Influenza viruses infect the epithelial cells of the respiratory mucosa, to which they bind by their surface glycoprotein, hemagglutinin (HA), to the cell receptor, sialic acid [10]. Human influenza is caused by influenza virus types A and B [12]. There are 18 HA subtypes of influenza type A [13]. On the basis of HA antigenic differences and phylogenetic sequence relatedness, influenza A viruses are also classified into group 1 (H1, H2, H5, H6, H8, H9, H11, H12, H13, H16, H17, and H18) and group 2 (H3, H4, H7, H10, H14, and H15) [14]. Influenza B viruses are not divided into subclasses, but into two major phylogenetic lineages, B/Victoria/2/87 and B/Yamagata/16/88, respectively termed the Victoria and Yamagata lineages in the 1980s [15,16,17]. Currently, human influenza is caused by two circulating influenza A viruses, H1N1 and H3N2, and influenza B virus, which lead to seasonal epidemics and occasional pandemics. HA is the major target of the host humoral immune response and vaccination [18].

HA is composed of a globular head domain (HA1), containing the receptor-binding site (RBS), and a stalk domain (HA2), containing the transmembrane domain (Figure 1A) [19,20]. In the 1980s, five antigenic sites A–E (A–D, Webster and Laver; E. Skehel et al.) have been identified on the HA1 surface of the H3N2 influenza virus [21,22,23]. Wiley and colleagues compiled a directory of amino acids in each of the antigenic sites and characterized the structural features and physical boundaries of each site [24]. Antigenic site B significantly overlaps with the RBS [24,25]. A recent report showed that antigenic site B has been immunodominant over site A in recently circulating H3N2 viruses [21]. Therefore, it was suggested that genetic mutations in antigenic site B drive the antigenic drift that facilitates immune evasion.

*N*-glycosylation of HA1 also helps the influenza virus to escape the host immune system [25,26]. Antibodies have limited access to major antigenic sites shielded by N-glycans on the HA1 domain. It has been suggested that the oligosaccharide chains accumulate in antigenic sites and contribute to immune evasion [23,27,28].

Influenza B viruses generally circulate in humans, while influenza A viruses can infect other species [29]. The antigenicity of HA is different in influenza A and B viruses [30]. Therefore, it is difficult to protect against influenza B virus infection, even with a universal vaccine or bnAbs against influenza A viruses [8,31].

### 2.2. Broadly Neutralizing Monoclonal Antibodies (bnAbs) Against Influenza Virus

The globular head domain HA1 contains major antigenic sites that are highly immunogenic and are able to elicit high serum antibody titers, and undergo antigenic variation [32,33,34,35]. The antibodies against RBS in HA1 can inhibit the attachment of the influenza virus to host cell receptors [18,35]. Mutations in this region would allow the virus to escape the host immune response [35]. However, a conserved RBS epitope, which is a potential target for a universal vaccine or bnAbs, was recently identified [36,37,38,39,40].

HA2, the stalk domain, also contains a highly conserved region, the fusion peptide, which is a possible therapeutic target (Figure 1A) [33,34]. The fusion peptide mediates the attachment of the viral envelope to the endosomal host membrane via a pH-triggered conformational change of HA. Neutralizing antibodies bind to the fusion peptide and prevent the conformation change responsible for membrane fusion, thus inhibiting infection [14]. However, HA2 stalk-specific antibodies are rarely produced during natural infections or conventional vaccinations [14,32]. On the basis of the possibility that HA2 is masked by the membrane-distal portion of HA, Palese and colleagues proposed a unique vaccine, lacking the globular head, HA1 [32]. Other studies using human subjects have suggested that the preexisting memory B cells against HA1 epitopes predominate, inhibiting the induction of protective antibodies against HA2 upon revaccination with similar strains [41]. The development of universal vaccines is required to overcome these obstacles. 

Several bnAbs targeting the fusion peptide have been generated (e.g., FI6 [42] and MEDI8852 [43]) and have been shown to successfully neutralize multiple influenza A virus subtypes of group 1 and group 2 HAs (Figure 1A) [14,44]. In general, human antibodies responding to the HA2 stalk domain (e.g., CR6261 [45], F10 [46], 27F3 [47], and CR9114 [37]) are mainly encoded by the VH1-69 germline gene (Figure 1B) [43,46,47,48]. A striking feature of the VH1-69-encoded antibodies is that the conserved hydrophobic residues are exposed, particularly the phenylalanine at position 54 (F54), located at the tip of the heavy-chain complementary determining region 2 (HCDR2) loop (Figure 1B) [45,46,47]. It is common for antibodies to use the conserved hydrophobic regions to target the conserved hydrophobic residues of viral proteins [49]. The VH1-69 germline gene also encodes bnAbs that recognize HA1 [50] or the corresponding targets of other viruses. These bnAbs include gp120, a membrane protein of human immunodeficiency virus type 1 (HIV-1) [51], and the RBS of severe acute respiratory syndrome coronavirus (SARS-CoV) [52] and Middle East respiratory syndrome (MERS)-CoV [53]. Lanzavecchia and colleagues found functional redundancy of the accumulated mutations and F54 in the process of affinity maturation (Figure 1B) [54]. Using genetic analysis of isolated B cell clones as previously described [54], the group also established MEDI8852, a bnAb that recognizes groups 1 and 2 of influenza A viruses [43]. They reconstructed the genealogical trees of the variable region from isolated B cell clones carrying the same germline gene segments. They then selected one clone, FY1, on the basis of its potency, breadth of reactivity to HA, and low number of somatic mutations, for affinity optimization with parsimonious mutagenesis [55], termed ex vivo affinity maturation. They generated MEDI8852 by modifying FY1 using ex vivo affinity maturation and succeeded in improving its affinity to H3 HA by 14-fold and that to H1 HA by 5-fold as compared to the affinity of the original. The clone contains a rare germline gene, *VH6-1*, but not a common germline gene, *VH1-69.* Their approach may be used to generate rare but potent bnAbs against the influenza virus.

### 2.3. Multidomain Antibody (MDAb) Engineering Using Nanobodies from Camelid Antibodies

Human IgGs consist of two identical heavy chains with one variable domain (V_H_), three constant domains (C_H_1, C_H_2, and C_H_3), and two identical light chains (Figure 2A) [56]. In 1993, Hamers et al. discovered that camelid antibodies have a totally different structure [57]. Camelid has the same structure as human IgGs, but IgG2 and IgG3 lack light chains and C_H_1 domains; nevertheless, IgG2 and IgG3 can bind a number of antigens (Figure 2B). Similar to a conventional V_H_, the camelid variable domain of a heavy chain-only antibody (V_H_H) (Figure 2C) consists of four constant framework regions (FR1–4) separated by three complementary determining region (CDR) loops (CDR1, 2, and 3) [58,59,60]. The CDR3 of V_H_H is more variable in both length and sequence than the other regions, and is similar to human V_H_ [56,60,61]. V_H_H forms a single-domain paratope with a molecular weight of approximately 15 kDa [58,62]. V_H_H is 10 times smaller than a whole IgG antibody (around 150 kDa) and half of a single-chain variable fragment (scFv) consist of a V_H_ and V_L_-linked domain and a linker oligopeptide of around 30 kDa [58,63,64]. Therefore, V_H_Hs are called “nanobodies” [64].

Compared with conventional IgG antibodies, V_H_Hs have several advantages. They are compact, and small V_H_Hs are remarkably stable under extreme conditions [60,62,65]. V_H_Hs also have a convex paratope, formed predominantly by the CDR3 loop, which allows binding to the clefts and pockets of the antigen to avoid recognition by conventional antibodies that have a dimeric concave or flat paratope (Figure 2C) [65,66]. They are cost-effective due to their high stability and ease of manufacture in microorganisms [67]. These advantages allow for the widespread use of V_H_Hs, including passive immunotherapy for infectious diseases [65].

One disadvantage of V_H_Hs is their rapid removal from circulation via renal filtration within a couple of hours after injection [58,68,69], with a threshold in the range of 40–60 kDa [70,71,72]. The short half-life limits the therapeutic application of V_H_Hs, and they require repeated administration. The half-life of conventional human IgG is approximately 3 weeks, due to the presence of a neonatal Fc receptor, FcRn [56,73]. Therefore, the fusion of a human Fc segment to a V_H_H will extend its half-life [58,62,65,74]. Enlarging the size of V_H_Hs from 15 kDa to 80 kDa also prolongs their retention in the body [75]. It has been reported that intranasal inoculation with V_H_H-Fc fusions against neuraminidase (NA), another membrane protein, succeeded in neutralizing a lethal influenza virus infection in mice [76].

Single-domain V_H_Hs are easily modified via genetic multimerization to form bispecific, bivalent, or tetravalent molecules to enhance function (Figure 2D) [77,78]. Bivalent MDAbs against tumor necrosis factor (TNF) molecules exhibited 100–500-fold greater binding and neutralizing ability than monovalent V_H_Hs [79]. The length of a linker sequence, such as the glycine–serine (GS) linker, affected the potency (16-fold) [80]. The linker of a bivalent MDAb contains up to several dozens of amino acids in order to avoid steric clashes with the antigens. A bispecific MDAb, which contains two V_H_Hs, each recognizing a different epitope, is more potent than a monomeric V_H_H [80]. An up to 10-fold greater potency can be obtained, depending on the relative position of V_H_H molecules.

Laursen et al. prepared several broadly neutralizing V_H_Hs against influenza A and B viruses by immunizing llamas with recombinant HA [8]. They isolated two neutralizing V_H_Hs against the influenza A virus (SD36 and SD38), and two against the influenza B virus (SD83 and SD84) (Table 1 and Figure 3). Next, they generated bivalent MDAbs with SD38 linked to SD36 and SD83 linked to SD84 by 18-, 38-, and 60-residue linkers. Although the length of the linker did not significantly affect the neutralizing activity, bivalent MDAbs were notably more potent at neutralizing influenza virus strains than monovalent V_H_Hs. Laursen et al. generated a tetravalent MDAb, MD2407, via a genetic fusion of SD38, SD36, SD83, and SD84, using 10-residue linkers (Figure 3). The linkers were too short to allow the V_H_Hs to bind the same HA molecule. In addition, they combined the tetravalent MDAb and human IgG1 Fc to obtain a tetravalent MDAb–Fc fusion protein (MD3606) (Figure 3). MD2407 and MD3606 neutralized all influenza A and B viruses tested, except for the H12 virus. One reagent provided broad protection against both influenza A and B viruses.

## 3. Recent Passive Immunization via Antibody–Gene Transfer

Several factors hinder the widespread use of antibody drugs, such as the necessity of weekly or biweekly infusions due to short half-life (approximately 20 days) [86], high production cost, laborious purification procedures, and quality control [87]. To address the above problems, a novel passive immunization method using plasmids or viral vectors encoding the antibody gene has been introduced. Long-term neutralizing antibodies are produced by the host cells following a single administration. In addition, gene-based antibody drugs are cost-effective in terms of production, purification, and administration [88,89].

Previously, for the first time, we induced over 10 μg/mL of the neutralizing anti-HA antibodies to BALB/c mice using electro-gene transfer with a plasmid encoding the neutralizing antibody gene [90]. The expression was stable for at least 70 days after inoculation. These potent and stable neutralizing antibodies provided long-lasting protection against influenza infection. We also administered passive immune treatment via hydrodynamic injection, which involves a rapid injection of a large volume of plasmid DNA solution into a mouse tail vein [91]. Neutralizing antibodies were detected within 4 hours of injection. We then successfully treated a lethal influenza virus infection 2 days after the challenge. Although we have not carried out the cross-protection experiments yet, another group has demonstrated broad cross-protection with plasmid DNA encoding two bnAbs that target influenza A and B viruses [92]. Several reports have described potent and stable cross-protection against influenza virus infection achieved via antibody–gene transfer with adeno-associated virus (AAV) vectors [89]. One study indicated that passive immunization via intranasal delivery of an AAV vector encoding bnAb FI6 could broadly protect against influenza virus infections [93]. Laursen et al. generated an AAV vector encoding a humanized tetravalent MDAb–Fc fusion protein (MD3606) (Figure 3) [8]. These authors demonstrated broad cross-protection via intranasal injection of mice with the AAV construct 7 days before the challenge with a lethal dose of H1N1, H3N2, or influenza B virus. However, AAV vector immunogenicity and rare gene toxicity due to the integration of DNA into the host genome are concerning [94]. Passive immunotherapy has a long history, dating back to the use of antisera against tetanus and diphtheria, first developed by Kitasato and Behring in the 19th century [95,96]. In the future, passive immunotherapy using antibody gene-encoding MDAbs, such as MD3606, could broadly protect against influenza virus infections.

## 4. A Universal Intranasal Vaccination against Influenza

### 4.1. Features of Intranasal Vaccination for Influenza Virus Infection

Effective antibody production depends upon the route of vaccine administration. Several types of vaccines are available, such as live attenuated influenza vaccines (LAIVs), inactivated viral vaccines, and subunit vaccines [97]. In general, LAIVs are administered intranasally, whereas current inactivated vaccines are administered subcutaneously or intramuscularly. Current intranasal LAIVs can minimize viral infection and induce the production of both secretory pIgA and IgG in the respiratory tract [98]. However, LAIVs are not approved for high-risk individuals, due to the transmission of the attenuated but live virus [98]. Conventional subcutaneous vaccines can mainly induce serum IgG generation but are not effective against the variant viruses [99]. Adjuvant-combined, nasal-inactivated vaccines, however, induce potent secretory pIgA and serum IgG production. Induced pIgA production provides broadly neutralizing activity against the influenza virus in the upper respiratory tract; that is, pIgAs can effectively protect against both homologous and heterologous influenza virus infections [99]. We first discuss previous research into intranasal vaccination. 

A study was conducted by Tamura et al. [98] to compare the efficacy of the intra-nasal and subcutaneous routes of administration of influenza vaccines. Vaccines were prepared against influenza virus strains A/Guizhou-X (H3N2), A/Fukuoka (H3N2), A/Sichuan (H3N2), A/PR8 (H1N1), and B/Ibaraki using the cholera toxin B subunit (CTB) as an adjuvant [100]. Mice were challenged with a non-lethal infection of A/Guizhou-X. All intranasal vaccinations except B/Ibaraki provided secretory IgA cross-reacting to A/Guizhou-X-HA (Table 2). Only homologous vaccination could induce the specific IgG to A/Guizhou-X-HA. The protective effect of the challenged virus reflected the titer of specific antibodies (Table 2). Vaccination with A/PR8, which is the other subtype from the challenged virus, provided low cross-protection, and vaccination with influenza B virus failed to induce cross-protection and induction of secretory IgA. No subcutaneous vaccinations induced the specific IgA, although they could induce a potent specific IgG reacting only to the homologous strain. Overall, intranasal vaccinations led to the production of more potent secretory IgA and cross-protection than conventional vaccinations.

It is also important to have a suitable adjuvant that is safe for human use in the clinical application of intranasal influenza vaccines. Several studies have been conducted into reducing side effects by the introduction of mutations in CTB [101] or complement component C3d [102]. Ichinohe et al. demonstrated another effective adjuvant, a synthetic double-stranded RNA (dsRNA) polyriboinosinic polyribocytidylic acid (poly (I:C)) for intranasal vaccination [103]. Clinical studies have also suggested that secretory pIgAs, generated in response to inactivated nasally administered vaccine, play a significant role in providing protection from heterologous influenza viruses [104,105]. An intranasally delivered inactivated vaccine could be valuable in promoting some of the favorable immune responses elicited by LAIVs, without the risk of live virus to immunocompromised individuals. In the next section, we focus on the mechanism of secretory pIgA protection against influenza virus infections.

### 4.2. Secretory pIgAs against Influenza Virus Infections

Among the isotypes, IgA is the principal immunoglobulin found on mucosal surfaces [106]. Characteristic features differentiate IgA from others, notably its quaternary structure [107]. While IgAs almost always exist in the monomeric form in human serum, those in the lamina propria of mucosal tissue covalently link with the heavy chains to the J chain to form polymers [108]. pIgA binds to the polymeric immunoglobulin receptor (pIgR) on the basolateral surfaces of mucosal epithelial cells via the J chain. The complex is transported across epithelial cells to the apical surface, where pIgR is cleaved, releasing a secretory component (SC), which remains associated with pIgA in the lumen. Secretory pIgA exists mainly in the dimeric form, with low levels of the tetrameric form [11,107,109]. Studies have been conducted to generate recombinant secretory pIgA, but the main focus was on the dimeric form [110,111,112].

Suzuki et al. showed that tetrameric IgA (tet-IgA) offers more protection against influenza A viruses in nasal mucosa than monomeric or dimeric forms [11]. To evaluate the polymerization of secretory pIgA, and the molecular mechanism of protection against influenza infection, several antibodies with identical variable regions are needed in monomer, dimer, and tetramer form. Saito et al. recently generated recombinant secretory pIgAs [113]. They produced recombinant monoclonal human tet-IgAs by co-expressing the heavy, light, and J chains along with SC in mammalian cells. They compared the reactivity and functionality of the generated bnAbs, including monomeric IgA, dimeric IgA, and tet-IgA. Compared to the other forms, recombinant tet-IgAs exhibited enhanced neutralizing activity against low- but not high-affinity virus strains. It has been suggested that tetramerization significantly enhances the neutralization breadth of IgA. Overall, to prevent an influenza outbreak, a safe vaccine should provide broad, long-term cross-protection. Compared with conventional vaccines, a nasally administered inactivated vaccine is expected to provide universal protection against influenza infections. 

## 5. Conclusions

We present an overview of novel passive immunotherapy using tetravalent MDAbs involving four V_H_Hs and tet-IgA induced by intranasal vaccination for influenza. Our conclusions include the following three points. First, antigen design based on physical features is important. Next, we discussed the conformation of the antibody for cross-protection. Finally, we focused on the technology of MDAb engineering.

Several conserved HA targets have been identified (e.g., fusion peptides in the stalk domain), and many bnAbs are proceeding to clinical trials [14]. There are also ongoing clinical trials of intranasal vaccinations with inducible neutralizing pIgAs against HA to achieve cross-protection [105]. However, it is difficult to obtain HA2 stalk-specific antibodies with natural infections or conventional vaccinations. Considering the highly biased usage of the hydrophobic *VH1-69* gene for antibodies against the HA2 stalk [41], the hydrophobic epitopes (e.g., HA2) might have the drawback; although the antibodies targeting the hydrophobic epitope have a tendency to broad reactivity, they are rarely induced by conventional vaccination. Antigen design based on physical features such as hydrophobicity, clefts, and pockets, is also essential to the development of bnAbs.

The larger secretory pIgAs induced by intranasal vaccination had high neutralizing efficacy against an influenza virus infection [11]. Recombinant technology producing tetrameric IgA (tet-IgA) could be also powerful tool, inducing passive immunization. Several studies have already succeeded in inducing potent neutralizing IgG antibodies using antibody–gene transfer with plasmid or adeno-associated virus (AAV) vectors [86]. Isolated monoclonal pentameric IgMs also broadly protect against the influenza B virus [114], although it is difficult to induce potent neutralizing polymeric IgMs via vaccination [11]. Monoclonal IgGs were demonstrated to spontaneously assemble into hexamers [115]. However, IgG hexamers have not been shown to effectively neutralize influenza. Further research is needed to understand the relationships between the quaternary structure and the neutralizing effects of antibodies.

One of the advantages of V_H_Hs is that they are compact, allowing easy genetic modification. Another is their stability under extreme conditions. They are a unique form of paratope consisting of a single domain. It would be valuable to identify the unique germline gene of V_H_H which is homologous to *VH1-69* in human IgG and obtain the sequence for ex vivo affinity maturation. Passive immunization with multivalent MDAbs based on V_H_H could control circulating influenza virus. If we succeeded in generating multivalent MDAb with several V_H_Hs against each influenza A virus (group 1 or group 2), or B virus, even one reagent would induce cross-protection for whole influenza subtypes. These strategies could help to prevent future influenza pandemics.

## Figures and Tables

**Figure 1 vaccines-08-00424-f001:**
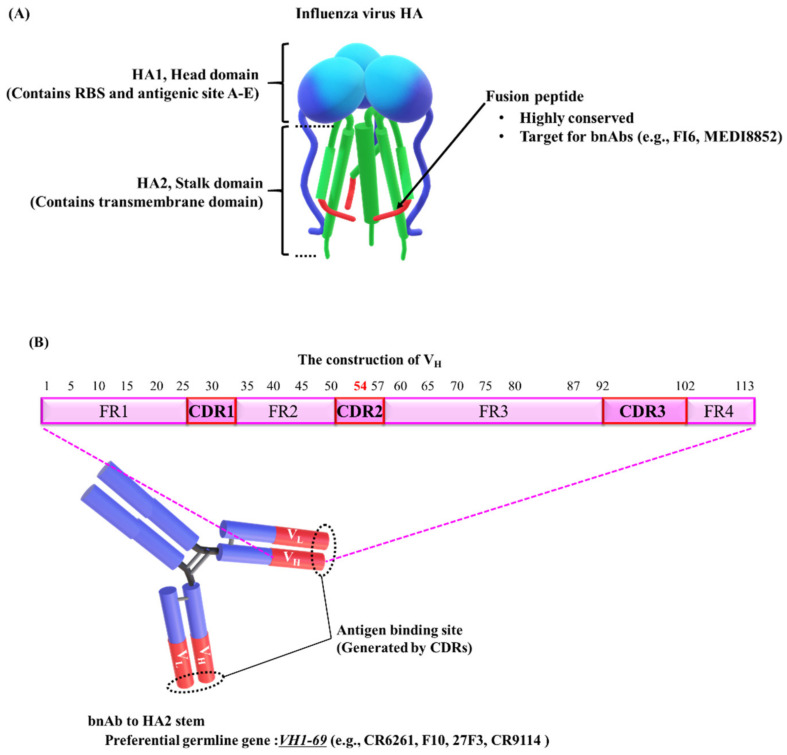
Schematic diagram of HA, the target for broadly neutralizing antibodies (bnAbs). (**A**) The model shows hemagglutinin (HA) structure (prefusion). HA is composed of a globular head domain (HA1) and a stalk domain (HA2) with a highly conserved sequence, the fusion peptide. The HA structure model was obtained from [81,82,83,84]. (**B**) The VH1-69 germline gene generally dominates the human bnAb response to the HA2 stalk domain. The variable domain of heavy chains (V_H_) consists of four framework regions (FR1-4) and three complementary determining regions (CDR1-3). The phenylalanine at position 54 (F54) is conserved in VH1-69 and is required for initial development of most VH1-69 antibodies. However, in the process of affinity maturation, F54 and accumulated mutations are functionally redundant. The numbers indicate the residue positions [54].

**Figure 2 vaccines-08-00424-f002:**
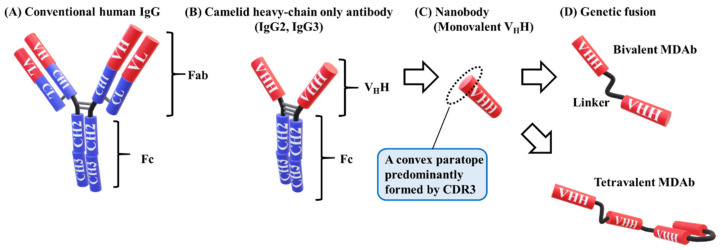
Model of human and camelid IgG. (**A**) Conventional human IgG consists of two identical heavy chains with one variable domain (V_H_), three constant domains (C_H_1, C_H_2, and C_H_3), and two identical light chains with one variable domain (V_L_) and one constant domain (C_L_). Fab contains the antigen-binding sites, and Fc is involved in effector functions. (**B**) The structure of camelid IgG1 (not shown) is similar to that of human IgG. Camelid IgG2 and IgG3 lack light chains and C_H_1 domains. The hinge region of IgG2 is longer than that of IgG3 (the model is representative) [85]. (**C**) V_H_H, known as a nanobody, consists of a variable domain of camelid IgG (IgG2 or IgG3) and forms a paratope with a single domain. The molecular weight is approximately 15 kDa. A convex paratope is predominantly formed by the CDR3 loop to bind to clefts or pockets of the antigen. (**D**) V_H_H is easily modified via genetic multimerization to a bivalent or tetravalent multidomain antibody (MDAb).

**Figure 3 vaccines-08-00424-f003:**
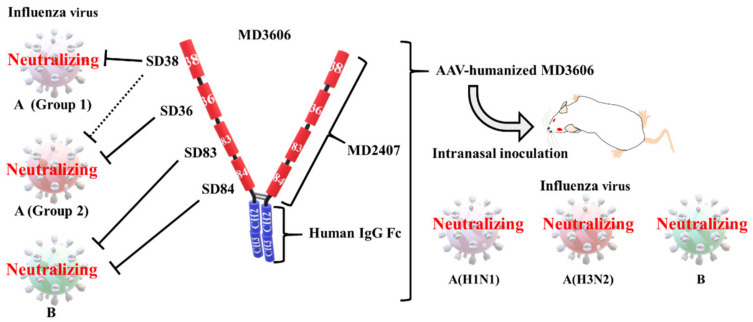
Cross-protection with a heavy chain only antibody (V_H_H)–Fc fusion construct. All four V_H_Hs were obtained by immunizing llamas with recombinant hemagglutinin (HA). V_H_H (SD38) can neutralize group 1 influenza A viruses and weakly neutralize some group 2 viruses, V_H_H (SD36) neutralizes group 2 influenza A viruses, and V_H_H (SD83 and SD84) neutralizes influenza B viruses [8]. A tetravalent multidomain antibody (MDAb; MD2407) was generated by connecting V_H_Hs with (GGGGS)_2_-linkers. An MDAb–Fc fusion (MD3606) was generated by attaching MD2407 to the human IgG Fc region. MD2407 and MD3606 neutralized influenza A and B viruses in vitro. Intranasal injection with an AAV vector encoding a humanized MD3606, administered 7 days before exposure, provided broad cross-protection against a lethal dose of H1N1, H3N2, and influenza B virus.

**Table 1 vaccines-08-00424-t001:** Neutralizing potency of isolated V_H_Hs against the influenza A or B viruses [8]. Neutralizing titer was determined by in vitro microtiter assay.

V_H_Hs	High Neutralizing Potency	Low or No Neutralizing Potency
SD36	Influenza A group 2 (H3, H4, H7, and H10)	Influenza A group 1 (H1, H2, and H5)
SD38	Influenza A group 1 (H1, H2, and H5)	Influenza A group 2 (H3 *, H4, H7 *, and H10 *)
SD83	Influenza B (Yamagata and Victoria linage)	-
SD84	Influenza B (Yamagata and Victoria linage)	-

* Lower neutralizing potency.

**Table 2 vaccines-08-00424-t002:** Comparison table of the efficacy between intranasal (i.n.) and subcutaneous (s.c.) vaccines for influenza [98]. N.S., not significant. HA (A/Guizhou-X)-Reactive IgA in nasal wash; +, −5 ng/mouse; ++, 5–10 ng/mouse; +++, 10–15 ng/mouse. HA (A/Guizhou-X)-Reactive IgG in nasal wash; ++++, 20–25 ng/mouse; +++++, 40–45 ng/mouse. EID_50_, 50% egg infectious dose. Protection against A/Guizhou-X in nasal wash; N.S., approximately 1 × 10^4^ EID_50_; ++, approximately 1 × 10^2^ EID_50_; +++, approximately 1 × 10 EID_50_.

Vaccine Strain	Subtype	Route	HA (A/Guizhou-X)-Reactive IgA	HA (A/Guizhou-X)-Reactive IgG	Protection against A/Guizhou-X
A/Guizhou-X	H3N2	i.n.	+++	++++	+++
A/Fukuoka	H3N2	i.n.	++	N.S.	+++
A/Sichuan	H3N2	i.n.	++	N.S.	+++
A/PR8	H1N1	i.n.	+	N.S.	++
B/Ibaraki	-	i.n.	N.S.	N.S.	N.S.
A/Guizhou-X	H3N2	s.c.	N.S.	+++++	++
A/Fukuoka	H3N2	s.c.	N.S.	N.S.	N.S.
A/Sichuan	H3N2	s.c.	N.S.	N.S.	N.S.
A/PR8	H1N1	s.c.	N.S.	N.S.	N.S.
B/Ibaraki	-	s.c.	N.S.	N.S.	N.S.

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
