# Peer review of "Broadly Neutralizing Antibodies for Influenza: Passive Immunotherapy and Intranasal Vaccination"

_vaccines, 2020, doi:10.3390/vaccines8030424_

Round 1
Reviewer 1 Report
The review paper by Biswas et al. focuses on the antibody engineering for cross-protection against influenza virus infection. In the Introduction the authors discuss the necessity of the design of broadly neutralizing antibodies (bnAbs) due to the influenza antigenic drift and antigenic shift of hemagglutinin and neuraminidase - main surface antigens. The second part contains a brief overview of a universal intranasal vaccination against influenza. Intranasal administration is advantageous over subcutaneous or intramuscular since it can induce IgA production in the upper respiratory tract. This is important because IgA is the principle immunoglobin of mucosal surfaces. The rest of the article describes passive immunization strategies for cross protection against influenza. The authors describe five antigenic sites of the HA1 in a wider context of head/stalk hemagglutin structure. Section 3.3 contains the description of nanobodies from camelid antibodies which can be reduced to approx. 15kDa and show amazing thermal stability. Next, multidomain antibody engineering is discussed, followed by the description of current strategies of antibody-gene transfer. The article ends with some perspectives on overcoming current COVID-19 outbreak and conclusions. In the opinion of the reviewer, the article is well written and the content is well balanced. The topic is properly introduced by providing condensed information on influenza virus features. Therefore the article may reach wider audience.
Author Response
Response to Reviewer 1 Comments
Comment: The review paper by Biswas et al. focuses on the antibody engineering for cross-protection against influenza virus infection. In the Introduction the authors discuss the necessity of the design of broadly neutralizing antibodies (bnAbs) due to the influenza antigenic drift and antigenic shift of hemagglutinin and neuraminidase - main surface antigens. The second part contains a brief overview of a universal intranasal vaccination against influenza. Intranasal administration is advantageous over subcutaneous or intramuscular since it can induce IgA production in the upper respiratory tract. This is important because IgA is the principle immunoglobin of mucosal surfaces. The rest of the article describes passive immunization strategies for cross protection against influenza. The authors describe five antigenic sites of the HA1 in a wider context of head/stalk hemagglutin structure. Section 3.3 contains the description of nanobodies from camelid antibodies which can be reduced to approx. 15kDa and show amazing thermal stability. Next, multidomain antibody engineering is discussed, followed by the description of current strategies of antibody-gene transfer. The article ends with some perspectives on overcoming current COVID-19 outbreak and conclusions. In the opinion of the reviewer, the article is well written and the content is well balanced. The topic is properly introduced by providing condensed information on influenza virus features. Therefore, the article may reach wider audience.
Response: We are grateful for the insightful and inspiring comments of the reviewer. Based on the comments, we combined the introductory texts about influenza to simplify our manuscript. Then, we modified the order of the topics as follows: the first topic is about “Antibody Engineering” in Section 2 and “Recent Passive Immunization” in Section 3, followed by “A Universal Intranasal Vaccination
Against Influenza” in Section 4. According to other reviewers’ comments, we also deleted the topic about SARS-CoV-2 to avoid confusion. Our revisions are clearly highlighted using the "Track Changes" function in Microsoft Word, and the reviewer’s number is attached. Thank you in advance for your kind consideration of our manuscript.
Reviewer 2 Report
Brief Summary
This paper is a review of recent developments in antibody engineering towards the goal of a broadly neutralizing universal influenza vaccine. The review covers aspects of influenza biology, antibody structure and function, and the pursuit of antibody-based strategies to protect against influenza virus disease. These strategies include induction of IgA, induction of broadly neutralizing antibodies, and therapy with camelid-derived nanobodies. Importantly, the review also highlights technology for elicitation of passive immunity by antibody gene transfer, a topic not covered by the most recent similar review.
Broad Comments
Overall, this paper appears to be a timely review as I could find no similar review of influenza virus-specific antibody engineering published in the past two years. However, the review from 2018 (https://doi.org/10.1155/2018/9747549) is fairly thorough and covered some of the same topics. It would be good to cite that review. Furthermore, it would improve the paper to pivot it away from COVID-2 (see comments below) and rather re-focus the paper as an update to the Chaisri et al. review. on the status of these technologies since 2018. The extent of the content and references relating to influenza vaccines appears to be appropriate.
The paper would benefit from a minor amount of copyediting throughout to improve clarity, readability and sentence structure. Some, but not all, of the English language issues are noted in the Minor Comments.
Major Criticisms:
The focus of this paper is on antibody engineering for cross-protection against the rapidly changing influenza HA glycoprotein. The SARS-CoV-2 mutation rate is very slow, especially as compared to influenza virus. Therefore, there is not a need for a broadly neutralizing antibody to combat SARS-CoV-2. I would recommend removing any reference to SARS -CoV-2, as it does not add to the paper. If the authors insist on referencing SARS-CoV-2 due to the current pandemic, it ought to only be in the context of how effective influenza therapeutics have the potential to improve public health outcomes in the scenario of a simultaneous influenza and SARS-CoV-2 epidemic. The review would be greatly improved by refocusing the entirety of the paper to influenza rather than trying to broaden the scope to cover SARS-CoV-2.
The paper would benefit from reorganization. Introductory text about influenza (Sections 2.1 & 3.1) should be combined at the beginning of the article, while sections about antibody strategies (Sections 2.3,& 3.2, 3.3) can be placed in subsequent sections.
Specific Comments
Line 38: “However, current vaccines are not universal” should be modified to “not all universal” or similar, as some current vaccines are still universal.
Line 47: The head domain contains more than one neutralizing epitope.
Section 2.2 could use improved clarity. It would help to add a sentence to the first paragraph of this section which states to the point of this section, e.g.: “An intranasally delivered, yet inactivated vaccine, may be a strategy to promote some of the favorable immune responses elicited by LAIVs without the risk to immunocompromised individuals.”
Line 118: “need” should be “needed”
I would recommend that in section 3.2, where antibodies against the fusion peptide are referenced, that you draw attention to the hypothesis of Andrews et al (your reference 61) that the hydrophobicity of the fusion peptide might mean that antibodies targeting this epitope have a tendency to be polyreactive and therefore under negative selection. I think this is an important point of consideration for vaccine strategies.
Line 196 references Figure 1 but the referenced position 54 is not labelled in Figure 1.
Line 247: “prolong” should be “prolonged”
Line 300: Include which animal model was used for this study.
Author Response
Response to Reviewer 2 Comments
Comment: (Brief Summary) This paper is a review of recent developments in antibody engineering towards the goal of a broadly neutralizing universal influenza vaccine. The review covers aspects of influenza biology, antibody structure and function, and the pursuit of antibody-based strategies to protect against influenza virus disease. These strategies include induction of IgA, induction of broadly neutralizing antibodies, and therapy with camelid-derived nanobodies. Importantly, the review also highlights technology for elicitation of passive immunity by antibody gene transfer, a topic not covered by the most recent similar review.
(Broad Comments) Overall, this paper appears to be a timely review as I could find no similar review of influenza virus-specific antibody engineering published in the past two years. However, the review from 2018 (https://doi.org/10.1155/2018/9747549) is thorough and covered some of the same topics. It would be good to cite that review. Furthermore, it would improve the paper to pivot it away from COVID-2 (see comments below) and rather re-focus the paper as an update to the Chaisri et al. review. on the status of these technologies since 2018. The extent of the content and references relating to influenza vaccines appears to be appropriate. The paper would benefit from a minor amount of copyediting throughout to improve clarity, readability, and sentence structure. Some, but not all, of the English language issues are noted in the Minor Comments.
Response: We are grateful for the insightful and inspiring comments of the reviewer. We have revised the manuscript according to the reviewer’s comments and responded to them point by point. According to the reviewer’s suggestion, we combined the introductory texts about influenza viruses. Then, we found that the order of some sections needed modified (Please see comment in Response 1.). We also cite the useful reference from the reviewer’s recommendation in Section 2.3 on pages 4-5, lines 178-181. We also deleted the topic regarding SARS-CoV-2 to avoid confusion. We feel that the revised manuscript is a suitable response to the comments and is significantly improved over the initial submission. Our revised manuscript was also reviewed by an English editing service (Editage, www.editage.jp). Our revisions are clearly highlighted using the "Track Changes" function in Microsoft Word, and the reviewer’s number is attached. Thank you in advance for your kind consideration of our manuscript.
Point 1: (Major Criticisms) The focus of this paper is on antibody engineering for cross-protection against the rapidly changing influenza HA glycoprotein. The SARS-CoV-2 mutation rate is very slow, especially as compared to influenza virus. Therefore, there is not a need for a broadly neutralizing antibody to combat SARS-CoV-2. I would recommend removing any reference to SARS -CoV-2, as it does not add to the paper. If the authors insist on referencing SARS-CoV-2 due to the current pandemic, it ought to only be in the context of how effective influenza therapeutics have the potential to improve public health outcomes in the scenario of a simultaneous influenza and SARS-CoV-2 epidemic. The review would be greatly improved by refocusing the entirety of the paper to influenza rather than trying to broaden the scope to cover SARS-CoV-2. 

The paper would benefit from reorganization. Introductory text about influenza (Sections 2.1 & 3.1) should be combined at the beginning of the article, while sections about antibody strategies (Sections 2.3,& 3.2, 3.3) can be placed in subsequent sections.
Response 1: We appreciate the reviewer’s comment. As the reviewer’s comment describes, the SARS-CoV-2 mutation rate is very slow. Therefore, we concluded that this topic does not fit the focus of our manuscript; therefore, we removed the topic regarding SARS-CoV-2. We also combined the introductory texts about influenza viruses in Section 2.1 on page 2. Then, we found that the introductory texts were more related to the topic,“Antibody Engineering.” Therefore, we changed the order as follows; the first topic involves “Antibody Engineering” in Section 2 and “Recent Passive Immunization” in Section 3, followed by “A Universal Intranasal Vaccination against Influenza” in Section 4.
Point 2: Line 38: “However, current vaccines are not universal” should be modified to “not all universal” or similar, as some current vaccines are still universal.
Response 2: We appreciate the reviewer’s comment. We changed this to “not all universal” on page 2, line 51.
Point 3: Line 47: The head domain contains more than one neutralizing epitope.
Response 3: We appreciate the reviewer’s comment, and we deleted the sentence on page 2, line 60.
Point 4: Section 2.2 could use improved clarity. It would help to add a sentence to the first paragraph of this section which states to the point of this section, e.g.: “An intranasally delivered, yet inactivated vaccine, may be a strategy to promote some of the favorable immune responses elicited by LAIVs without the risk to immunocompromised individuals.”
Response 4: We appreciate the reviewer’s comment. We inserted your suggested sentence in Section 4.1, page 9, line 351-353 for clarity. Moreover, we also improved the information regarding intranasal vaccinations for cross-protection with some evidence including antibody titer and protection effects compared with conventional subcutaneous vaccinations.
Point 5: Line 118: “need” should be “needed”
Response 5: We apologize for the mistake. We corrected this on page 9, line 369.
Point 6: I would recommend that in section 3.2, where antibodies against the fusion peptide are referenced, that you draw attention to the hypothesis of Andrews et al (your reference 61) that the hydrophobicity of the fusion peptide might mean that antibodies targeting this epitope have a tendency to be polyreactive and therefore under negative selection. I think this is an important point of consideration for vaccine strategies.
Response 6: We appreciate the reviewer’s recommendation. We discussed the Andrews et al. hypothesis, as suggested, in the Conclusion section on page 10, lines 396-3401.
Point 7: Line 196 references Figure 1 but the referenced position 54 is not labelled in Figure 1.
Response 7: We apologize for this mistake. We removed the reference to Figure 1 in the sentence on page 4, line 161, as per the reviewer’s recommendation.
Point 8: Line 247: “prolong” should be “prolonged”
Response 8: We apologize for the mistake. We corrected this on page 5, line 213.
Point 9: Line 300: Include which animal model was used for this study.
Response 9: We appreciate the reviewer’s comment. We clarified the animal model to indicate “BALB/c mice” in the sentence on page 8, line 289.
Reviewer 3 Report
In the manuscript “Antibody Engineering for Cross-Protection against Influenza Virus Infection; Clues for Overcoming the Coronavirus (COVID-19) Outbreak”, the authors reviewed recent advances in influenza virus vaccine development, engineering of broadly neutralizing antibody (bnAb), prospect of passive vaccines with these bnAbs, as well as the implications for overcoming the Coronavirus (COVID-19) Outbreak.
In general, this review has covered many topics related to influenza virus vaccine and antibody discovery and engineering. However, on the other hand, this manuscript failed to provide a comprehensive review on each of these topics in depth.
Major concerns:
- In the section “A Universal Intranasal Vaccination against Influenza”, the authors highlighted the advance in the intranasal vaccination, suggesting it is “more effective cross-protection than conventional vaccination”. It would be more convincing to add more evidence and discussion to support this conclusion, such as the virus subtype used, how to produce the antigen, how to deliver the vaccine, the test data, antibody titer induced by the vaccination, protection effects, duration, and a comparison to conventional vaccine. Based on all these evidences, the authors can further discuss the mechanism why the intranasal vaccination is better than conventional vaccination against influenza virus.
- In the section “Passive Immunization for Cross Protection against Influenza Virus Infection”, the authors talked about the discovery of broadly neutralizing antibodies against influenza virus, camelid antibodies and passive immunization with vectorized Abs. This section contains too much background information about influenza virus HA and camelid antibodies, which are relevant but not essential to the topic. Please consider to reduce the volume of these descriptions. While the authors didn’t talk much about antibody engineering, although the title of this paper is “Antibody Engineering…”
- In the section “A Perspective on Overcoming the COVID-19 Outbreak”, I don’t see much interesting points on how to transfer the influenza virus research experience to overcoming the COVID-19 outbreak. If the authors would recommend bnAbs to target COVID-19 outbreak, there are more than a hundred anti-HIV bnAbs have been isolated.
Author Response
Response to Reviewer 3 Comments
Comment: In the manuscript “Antibody Engineering for Cross-Protection against Influenza Virus Infection; Clues for Overcoming the Coronavirus (COVID-19) Outbreak”, the authors reviewed recent advances in influenza virus vaccine development, engineering of broadly neutralizing antibody (bnAb), prospect of passive vaccines with these bnAbs, as well as the implications for overcoming the Coronavirus (COVID-19) Outbreak.
In general, this review has covered many topics related to influenza virus vaccine and antibody discovery and engineering. However, on the other hand, this manuscript failed to provide a comprehensive review on each of these topics in depth.
Response: We are grateful for the insightful and inspiring comments of the reviewer. We have revised our manuscript according to the reviewer’s comments and responded to them point by point. We considered the construct of our manuscript. In brief, we modified the title and modified the order of the sections; the first topic regards “Antibody Engineering” in Section 2 and “Recent Passive Immunization” in Section 3, followed by “A Universal Intranasal Vaccination Against Influenza” in Section 4. We have also added more information about intranasal vaccination (Please see our comments in Response 1.). We feel that the revised manuscript is a suitable response to the comments and is significantly improved over the initial submission. Our revised manuscript was also reviewed by an English editing service (Editage, www.editage.jp). Our revisions are clearly highlighted using the "Track Changes" function in Microsoft Word, and the reviewer’s number is attached. Thank you in advance for your kind consideration of our manuscript.
Point 1: In the section “A Universal Intranasal Vaccination against Influenza”, the authors highlighted the advance in the intranasal vaccination, suggesting it is “more effective cross-protection than conventional vaccination”. It would be more convincing to add more evidence and discussion to support this conclusion, such as the virus subtype used, how to produce the antigen, how to deliver the vaccine, the test data, antibody titer induced by the vaccination, protection effects, duration, and a comparison to conventional vaccine. Based on all these evidences, the authors can further discuss the mechanism why the intranasal vaccination is better than conventional vaccination against influenza virus.
Response 1: We appreciate the reviewer’s comment. We summarize the information regarding intranasal vaccinations for cross-protection with some evidence including antibody titer and protection effects compared with conventional subcutaneous vaccination in Section 4.1 on page 8. The information is summarized in Table 2 on page 9, line 342. We also deleted some background information regarding the influenza virus in the Introduction (Section 1) for simplicity on page 2.
Point 2: In the section “Passive Immunization for Cross Protection against Influenza Virus Infection”, the authors talked about the discovery of broadly neutralizing antibodies against influenza virus, camelid antibodies and passive immunization with vectorized Abs. This section contains too much background information about influenza virus HA and camelid antibodies, which are relevant but not essential to the topic. Please consider to reduce the volume of these descriptions. While the authors didn’t talk much about antibody engineering, although the title of this paper is “Antibody Engineering…”.
Response 2: We appreciate the reviewer’s comment. We combined the basic background information about the influenza virus from two sections (“Antibody Engineering” and “A Universal Intranasal Vaccination Against Influenza”) and simplified them on page 2. We also deleted some sentences regarding camelid antibodies for simplicity in Section 2.3 on page 4. Finally, we deleted ”Antibody Engineering” from the title and modified it to “Broadly Neutralizing Antibodies for Influenza: Passive Immunotherapy and Intranasal Vaccination ” to reflect our manuscript.
Point 3: In the section “A Perspective on Overcoming the COVID-19 Outbreak”, I don’t see much interesting points on how to transfer the influenza virus research experience to overcoming the COVID-19 outbreak. If the authors would recommend bnAbs to target COVID-19 outbreak, there are more than a hundred anti-HIV bnAbs have been isolated.
Response 3: We appreciate the reviewer’s comment and agree with the suggestion. Therefore, we deleted the topic regarding SARS-CoV-2 on pages 2 and 7.
Reviewer 4 Report
There is a constant mixture of new findings and/or arguments with those which had been exposed a couple of paragraphs earlier (or less!). The logics of the authors is much too often hard to follow. One feels that the authors did not have/take the time to reread their paper attentively and correct the mistakes, repeats, or redundancies it contains. And there are many (too many!) of them. Also, the repeated mixture between influenza virus and coronavirus too often makes the paper confusing! Finally, the conclusion of the paper is exactly 6 lines long (l.383-388)
Author Response
Response to Reviewer 4 Comments
Comment: There is a constant mixture of new findings and/or arguments with those which had been exposed a couple of paragraphs earlier (or less!). The logics of the authors is much too often hard to follow. One feels that the authors did not have/take the time to reread their paper attentively and correct the mistakes, repeats, or redundancies it contains. And there are many (too many!) of them. Also, the repeated mixture between influenza virus and coronavirus too often makes the paper confusing! Finally, the conclusion of the paper is exactly 6 lines long (l.383-388)
Response: We appreciate the reviewer’s assessment of our manuscript. According to the reviewer’s comments, we have revised our manuscript using the four points numbered below. Our revised manuscript was also reviewed by an English editing service (Editage, www.editage.jp). Our revisions are clearly highlighted using the "Track Changes" function in Microsoft Word, and the reviewer’s number is attached. Thank you in advance for your kind consideration of our manuscript.
- There is a constant mixture of new findings and/or arguments with those which had been exposed a couple of paragraphs earlier (or less!): We added more information about intranasal vaccination and provided more evidence, such as the virus subtype used, antibody titer induced by the vaccination, protection effects, and duration compared with conventional vaccines in Section 4.1 on page 8. The information is summarized in Table 2 on page 9, line 342.
- The logics of the authors is much too often hard to follow/ Also, the repeated mixture between influenza virus and coronavirus too often makes the paper confusing!: The section regarding SARS-CoV-2 is not relevant to the topic. Therefore, we deleted the topic about SARS-CoV-2 on pages 2 and 7.
- One feels that the authors did not have/take the time to reread their paper attentively and correct the mistakes, repeats, or redundancies it contains. And there are many (too many!) of them.: The comments suggested that the introduction about the influenza virus should be combined and simplified. Therefore, we combined basic background information about the influenza virus from two sections (“Antibody Engineering” and “A Universal Intranasal Vaccination Against Influenza”) in Section 2.1. We also deleted the redundant text on page 2, lines 58-64 and in Section 2.3 on page 4. We also apologize for our mistakes on page 9, line 369; page 4, line 161; and page 5, line 213.
- Finally, the conclusion of the paper is exactly 6 lines long (l.383-388): Our conclusions include the following three points. First, the antigen design based on physical features is important, as suggested by the highly biased germline gene (e.g. VH1-69) that is dependent on the features of the antigen. Next, we discussed the conformation of the antibody for cross-protection. Finally, we focused on the technology of MDAb engineering. We described that MDAb would be a powerful tool to manage influenza virus. (page 10, lines 383-414).
Reviewer 5 Report
When reading the title, I expect to be presented to the newest technology and newest information in antibody engineering, that means in vitro manipulation of antibodies.
I have a mixed impression of this review some parts are good and some are not so good.
Abstract. It doesn’t reflect the content of the review, with exception of the last sentence.
Introduction (section 1 and 2): can be shortened substantially. Much of the information is basic knowledge for anyone in this field.
Section 3 (passive immunization). There is lot of interesting information in this section. I think this section can be divided into two parts; antibody engineering and passive immunization. In the first part describe various technologies used to make antibodies, and the second part about experiences with passive immunization. Passive immunization was used under the Spanish flu and now with the Covid19 pandemic.
Section 4. This is an odd section, doesn’t fit the topic.
Section 5. This section is generic and doesn’t conclude what’s discussed this review.
I think this can be a nice review, but it needs to focus on the main topic.
Author Response
Response to Reviewer 5 Comments
Comment: When reading the title, I expect to be presented to the newest technology and newest information in antibody engineering, that means in vitro manipulation of antibodies.
I have a mixed impression of this review some parts are good and some are not so good.
Response: We are grateful for the insightful and inspiring comments of the reviewer. We have revised the manuscript according to the reviewer’s comments and responded to them point by point. In brief, we combined the introductory texts about the influenza virus to simplify our manuscript. Then, we modified the order of the topic and divided it into three topics (Please see response 2.). We also deleted the topic regarding SARS-CoV-2 to avoid confusion. We feel that the revised manuscript is a suitable response to the comments and is significantly improved over the initial submission. Our revised manuscript was also reviewed by an English editing service (Editage, www.editage.jp). Our revisions are clearly highlighted using the "Track Changes" function in Microsoft Word, and the reviewer’s number is attached. Thank you in advance for your kind consideration of our manuscript.
Point 1: Abstract. It doesn’t reflect the content of the review, with exception of the last sentence.
Response 1: We appreciate the reviewer’s comment. We modified the Abstract to reflect the main text on page 1, lines 13-34. We primarily introduce the tetravalent multidomain antibody MDAb-Fc that can broadly cross-protect against both the influenza A and B viruses and current observations of intranasal vaccination that can induce potent polymeric IgA.
Point 2: Introduction (section 1 and 2): can be shortened substantially. Much of the information is basic knowledge for anyone in this field.
Response 2: We appreciate the reviewer’s comment. We combined the introductory texts about the influenza virus in Section 2.1 on page 2. Then, we found that the introduction was more related to the topic, “Antibody Engineering.” Therefore, we modified the order as follows: the first topic is about “Antibody Engineering” in Section 2 and “Recent Passive Immunization” in Section 3, followed by “A Universal Intranasal Vaccination Against Influenza” in Section 4.
Point 3: Section 3 (passive immunization). There is lot of interesting information in this section. I think this section can be divided into two parts; antibody engineering and passive immunization. In the first part describe various technologies used to make antibodies, and the second part about experiences with passive immunization. Passive immunization was used under the Spanish flu and now with the Covid19 pandemic.
Response 3: We appreciate the reviewer’s comment. We divided passive immunization into two parts as the follows: “Antibody Engineering for Cross Protection Against Influenza Virus Infection” as Section 2 on page 2, line 88 and “Recent Passive Immunization via Antibody-Gene Transfer” as Section 3 on page 7, line 280.
Point 4: Section 4. This is an odd section, doesn’t fit the topic.
Response 4: We appreciate the reviewer’s comment. We deleted the topic about SARS-CoV-2 on page 7, lines 259-279.
Point 5: Section 5. This section is generic and doesn’t conclude what’s discussed this review.
I think this can be a nice review, but it needs to focus on the main topic.
Response 5: We appreciate the reviewer’s comment. Our conclusions include the following three points on page 10, lines 383-414. First, the antigen design based on physical features is important, as suggested by the highly biased germline gene (e.g., VH1-69) that is dependent on the features of the antigen. Next, we discussed the high structure of the antibody for cross-protection. Finally, we focused on the technology of MDAb engineering. We described that MDAb would be a powerful tool to manage the influenza virus.
Round 2
Reviewer 3 Report
The revised manuscript is more focused and concise. I agree to accept the current version for publishing.
Author Response
Thank you for your careful reading of our manuscript and for your kind consideration.
Reviewer 4 Report
This new version is much better written and easier to read than the first one. It was a good decision to take out all the references to Covid-19 and to rewrite a much better Introduction and slightly longer Conclusion. However, there still are much too many mistakes in the English writing of the text. Examples:
l. 69 :"..several bnAb against influenza virus have been isolated that demonstrate cross-protection.."
l. 74 : "...that the influenza viruses invade s the respiratory tract..."
l . 77-78 : "... that tetrameric IgAs have higher better neutralizing activity than monomeric or dimeric IgAs".
l . 92 : " The Influenza viruses infect epithelial cells of the respiratory mucosa to which they bind by their surface glycoprotein, hemagglutinin (HA)..."
Etc, etc... There are very many other sentences whose writing must be corrected! Please have an english-speaking person help you!
Fig 1: The figure is confusing and should be divided into two figures, Fig 1 for HA, Fig 2 for the bnAb. A clearer explanation on the germline gene is also needed. As is, the legend text and the writings in the Figure are totally incomprehensible!
The text which refers to fig 1 (l. 148-165) is pretty difficult to understand: what is an "ex-vivo affinity maturation"? and its" genetic analysis"? Where is "position 54" in the figure? What are "naturally ocurring bnAbs"? When and how are they induced?
Anyway, a bnAb cannot "neutralize multiple HA subtypes" (l. 149) it neutralizes the corresponding viruses!!
Fig 4: Do I understand correctly that the figure is withdrawn?
l.208: " and requires repeated administration..."
l. 210: "Therefore, the fusion of a human Fc segment to VhH will extend its half-life. Enlarging the size of VhHs from 15 to 80 kD also prolonged their retention in the body"
l. 325-329: A previous study was conducted by Tamura et al (105,107) to compare the efficacy of the intra-nasal and subcutaneous routes of administration of influenza vaccines. Vaccines were prepared against influenza virus strains A/Guizhou-X (H3N2), A/Fukuoka (H3N2), ..., and B/Ibaraki, using the cholera toxin B subunit (CTB) as an adjuvant. etc
Conclusion: The conclusion has happily been made slightly longer but it still remains too skinny. The results presented in the paper deserve to be better described and commented. But please write clear, shorter sentences! For instance, the paragraph l. 394- 401 is hardly understandable! As to the last paragraph, it has been written in pigeon english!!
Author Response
Response to Reviewer 4 Comments
Comment: This new version is much better written and easier to read than the first one. It was a good decision to take out all the references to Covid-19 and to rewrite a much better Introduction and slightly longer Conclusion. However, there still are much too many mistakes in the English writing of the text. Examples:
Response: We are grateful for the insightful comments and detailed suggestions of the reviewer. We have revised our manuscript according to the reviewer’s comments, and here we respond to them point by point.
We considered the structure of our manuscript. We divided Figure 1 into two figures; Figure 1A and Figure 1B. We clarified "ex-vivo affinity maturation" and "position 54" in the figure (Please see Responses 2 and 3). We also deleted the following words: “naturally occurring bnAbs" to avoid confusion. Finally, we modified and added our perspective in the Conclusions section (Please see Response 5). We feel that the revised manuscript addresses your concerns, and is significantly improved over the previous submission. Our revised manuscript was subjected to review by an English editing service (Editage, www.editage.jp). Our revisions are clearly highlighted using the "Track Changes" function in Microsoft Word, and the reviewer’s number is attached. Thank you in advance for your kind consideration of our manuscript.
Point 1: (However, there still are much too many mistakes in the English writing of the text. Examples: )
- 69 :"..several bnAb against influenza virus have been isolated that demonstrate cross-protection.."
- 74 : "...that the influenza viruses invade s the respiratory tract..."
l . 77-78 : "... that tetrameric IgAs have higher better neutralizing activity than monomeric or dimeric IgAs".
l . 92 : " The Influenza viruses infect epithelial cells of the respiratory mucosa to which they bind by their surface glycoprotein, hemagglutinin (HA)..."
Etc, etc... There are very many other sentences whose writing must be corrected! Please have an english-speaking person help you!
Anyway, a bnAb cannot "neutralize multiple HA subtypes" (l. 149) it neutralizes the corresponding viruses!!
l.208: " and requires repeated administration..."
- 210: "Therefore, the fusion of a human Fc segment to VhH will extend its half-life. Enlarging the size of VhHs from 15 to 80 kD also prolonged their retention in the body"
Response 1: We appreciate the reviewer’s detailed suggestions. We have made sure to incorporate them into the revised manuscript, along with other modifications to the language used, as needed.
Point 2: Fig 1: The figure is confusing and should be divided into two figures, Fig 1 for HA, Fig 2 for the bnAb. A clearer explanation on the germline gene is also needed. As is, the legend text and the writings in the Figure are totally incomprehensible!
Response 2: We appreciate the reviewer’s comment. We divided Figure 1 into two figures; Figure 1A indicates the structure of HA, and Figure 1B describes the details of the preferential germline gene VH1-69. We also added information about the phenylalanine at position 54 (F54) in Figure 1B. We include the two figures as sub-figures in Figure 1, because they are related.
Point 3: The text which refers to fig 1 (l. 148-165) is pretty difficult to understand: what is an "ex-vivo affinity maturation"? and its" genetic analysis"? Where is "position 54" in the figure? What are "naturally ocurring bnAbs"? When and how are they induced?
Response 3: We appreciate the reviewer’s comment. We added more information about the generation of MEDI8852, including ex-vivo affinity maturation on page 4, lines 165–170. Ex-vivo affinity maturation is optimization of affinity via parsimonious mutagenesis. It is a computer-assisted method for oligo deoxyribonucleotide-directed scanning mutagenesis. We also deleted "naturally occurring bnAbs" and modified the sentence on page 3, line 110, for clarity.
Point 4: Fig 4: Do I understand correctly that the figure is withdrawn?
Response 4: Yes. We completely deleted the information about COVID-19.
Point 5:l. 325-329: A previous study was conducted by Tamura et al (105,107) to compare the efficacy of the intra-nasal and subcutaneous routes of administration of influenza vaccines. Vaccines were prepared against influenza virus strains A/Guizhou-X (H3N2), A/Fukuoka (H3N2), ..., and B/Ibaraki, using the cholera toxin B subunit (CTB) as an adjuvant. etc
Response 5: We appreciate the reviewer’s modification. We have modified the sentence to reflected this wording. We also modified the sentences on page 7, lines 340–351 for clarity.
Point 5: Conclusion: The conclusion has happily been made slightly longer but it still remains too skinny. The results presented in the paper deserve to be better described and commented. But please write clear, shorter sentences! For instance, the paragraph l. 394- 401 is hardly understandable! As to the last paragraph, it has been written in pigeon english!!
Response 5: We appreciate the reviewer’s comment. We modified the sentence on page 9, lines 404–410 to clarify its meaning. We have also added our summary and perspective into each paragraph. In the last paragraph, we have edited the English extensively.

Reviewer 5 Report
None
Author Response

(The authors gave the same response as above.)
